# *Osteomeles schwerinae* Extract and Its Major Compounds Inhibit Methylglyoxal-Induced Apoptosis in Human Retinal Pigment Epithelial Cells

**DOI:** 10.3390/molecules25112605

**Published:** 2020-06-03

**Authors:** Bo-Jeong Pyun, Young Sook Kim, Ik Soo Lee, Dong Ho Jung, Joo-Hwan Kim, Jin Sook Kim

**Affiliations:** 1Herbal Medicine Division, Korea Institute of Oriental Medicine, 1672 Yuseongdae-ro, Yuseong-gu, Daejeon 34054, Korea; bjpyun@kiom.re.kr (B.-J.P.); jdh9636@kiom.re.kr (D.H.J.); 2Research Infrastructure Team, Herbal Medicine Division, Korea Institute of Oriental Medicine, Daejeon 34054, Korea; knifer48@kiom.re.kr; 3Department of Life Science, Gachon University, Seongnam, Kyonggi-do 13120, Korea; kimjh2009@gachon.ac.kr

**Keywords:** *Osteomeles schwerinae*, diabetic retinopathy, methylglyoxal, apoptosis, human retinal pigment epithelial cells

## Abstract

The accumulation and formation of advanced glycation end products (AGEs) are related to diabetes and age-related disease. *Osteomeles schwerinae* C. K. Schneid. (Rosaceae, OSSC) is used traditionally for the treatment of various diseases in Asia. Previous studies have shown that OSSC elicits preventive effects in an in vivo model of diabetes. This study was to evaluate the antiapoptotic effects of dried leaves and twigs of OSSC extract and its major compounds in ARPE-19 cells—spontaneously arising human retinal pigment epithelial cells—under diabetic conditions. To examine the effects of an OSSC extract and its active compounds (acetylvitexin, hyperoside and quercitrin) on apoptosis in methylglyoxal (MG, the active precursor in the formation of AGEs)-treated ARPE-19 cells and the mechanism by which these effects occur, apoptosis was measured using flow cytometry analysis. Protein expression levels of phospho-p53 (p-p53), Bax and Bcl-2 were determined by western blot analyses. The OSSC extract inhibited apoptosis in MG-treated ARPE-19 cells in a dose-dependent manner. The major compounds also reduced the rate of apoptosis. Both the extract and major compounds also inhibited the expression of p-p53 and Bax and increased the levels of Bcl-2 that had been previously reduced by MG treatment. The OSSC extract (0.1 μg/mL) and its major compounds (0.01 μM) attenuated apoptosis in ARPE-19 cells under toxic diabetic conditions by downregulating of expression of p-p53 and Bax. OSSC may serve as an alternative therapy to retard the development of diabetic retinopathy.

## 1. Introduction

The accumulation of advanced glycation end products (AGEs) is accelerated under conditions of chronic hyperglycemia and many age-related diseases [1]. Retinal pigment epithelial cells (ARPE-19 cells) are critical for the maintenance and survival of photoreceptors, as well as for the apoptosis of the RPE (retinal pigment epithelium) related to oxidative stress, inflammation and AGEs [2,3,4]. Methylglyoxal (MG), a highly reactive dicarbonyl compound, is an AGE precursor whose levels have been shown to increase in experimental and clinical diabetes [1]. Elevated MG levels have been shown to induce reactive oxygen species (ROS) production and apoptosis in various cell types [5,6,7]. Current research is focused on therapeutic and nutritional interventions to prevent apoptosis in diabetic retinopathy (DR) [8].

*Osteomeles schwerinae* C. K. Schneid. (OSSC), belonging to the family Rosaceae, is a perennial evergreen shrub, native to China that is harvested in the wild and used for local food [9]. Furthermore, this plant that has been used as a traditional Chinese medicine for the treatment of laryngopharyngitis, diarrhea, dysentery, folliculitis and hyperglycemia [10,11]. Previous studies have shown that OSSC ameliorates retinal endothelial cell apoptosis via the regulation of AGEs accumulation in the spontaneously diabetic torii rats [6,12]. Hyperoside, a compound of OSSC extract, has demonstrated significant inhibition of aldose reductase [13], the key enzyme in the polyol pathway during the pathogenesis of diabetic cataracts [14]. Furthermore, herbal extracts and their major compounds have been used for the treatment of diabetes and diabetic complications [15].

Here, we investigated whether an OSSC extract and its maker compounds could inhibit apoptosis in ARPE-19 cells treated with MG, thereby demonstrating their potential use to prevent the development of DR by inhibiting apoptosis in cultured RPE under toxic diabetic conditions.

## 2. Results

### 2.1. Effects of the OSSC Extract and Its Major Compounds on the Viability of MG-Treated Cells

After treatment with 500 μM MG for 24 h, cell viability was not significantly changed. However, at a concentration of 1000 μM, cell viability was significantly decreased to 65% (Figure 1A). FITC-conjugated annexin V and PI were used to monitor the progression of apoptosis. The percentage of early apoptotic (annexin V positive /PI negative) and late apoptotic/necrotic cells (annexin V positive-negative /PI positive) were determined. As shown in Figure 1B, the percentage of apoptotic cells was significantly increased from 6.11% (control) to 99.68% (at 5 mM MG). MG increased the rate of apoptosis in a concentration-dependent manner.

In a previous study, the purities of maker compounds of OSSC extract were analyzed by HPLC [12]. Here, we evaluated whether the extract and its maker compounds could affect cell viability in MG-treated ARPE-19 cells. As shown in Figure 1C, MG inhibited cellular proliferation, and pretreatment with the OSSC extract and its maker compounds significantly attenuated this inhibitory effect.

### 2.2 Effects of the OSSC Extract and Its Maker Compounds on MG-Induced Apoptosis

As shown in Figure 2A,B, MG increased the rate of apoptosis up to 30%; the OSSC extract and each maker compound attenuated the MG-induced increase in apoptosis. The effects of hyperoside were indistinguishable from the normal control. To assess the effects of the OSSC extract and its maker compounds on apoptosis-related factors involved in apoptosis, we examined the expression levels of p-p53, Bax and Bcl-2 by western blot analysis. MG induced a 2.3- and 1.4-fold increase in the expression of p-p53 and Bax, whereas the OSSC extract and its major compounds reduced the levels of these factors. Bcl-2 levels were reduced 0.8-fold after MG treatment, whereas the OSSC extract and its maker compounds increased Bcl-2 levels (Figure 2C).

## 3. Discussion

Herbal extracts and natural compounds have traditionally been used to treat diabetes and diabetic complications [8,12,16,17]. Here, an OSSC extract and its maker compounds were tested for their inhibitory effects on apoptosis and its related mechanisms under toxic diabetic conditions in ARPE-19 cells. The results showed that the OSSC extract and its maker compounds significantly inhibited apoptosis under diabetic conditions. MG-induced p53 phosphorylation and Bax-1 expression were reduced by OSSC treatment of ARPE-19 cells. Bcl-2 expression that had been attenuated by MG was increased by OSSC treatment in ARPE-19 cells.

Diabetic retinopathy gives rise to apoptosis in several retinal cell types, especially retinal pericytes and ARPE-19 cells [18]. The levels of several factors, such as glucose, ROS and AGEs, are elevated in diabetes and accelerate cellular injury and apoptosis [19,20]. Levels of MG are increased under diabetic conditions via a glucose-related mechanism and modification of lysine and arginine residues. Strategies to treat or prevent diabetic complications with medicinal herbs have focused on the inhibition of AGEs formation or its downstream pathways. The three maker compounds of the OSSC extract are 2’-*O*-acetylvitexin, hyperoside and quercitrin [6,21]; 2’-*O*-acetylvitexin was first isolated and characterized in *Trollius* (a genus of about 30 species of flowering plants), but no data previously exist with regard to its activity [22]. This study reveals the antiapoptotic activity of acetylvitexin in ARPE-19 cells under toxic diabetic conditions for the first time (Figure 2). Hyperoside is the predominant phenolic compound in the OSSC extract and inhibits oxidative stress via the activation of heme oxygenase-1 (HO-1) [23]. Quercitrin has been shown to elicit beneficial effects on diabetes, especially the inhibition of both aldose reductase activity and AGEs formation and the reduction of blood glucose levels [24,25]. These results suggest that the OSSC extract and its maker compounds may prevent the apoptosis of RPE cells under diabetic conditions.

## 4. Materials and Methods

### 4.1. Preparation of the OSSC Extract and Its Major Compounds

*O. schwerinae* was collected from Shrubs (2000 m), Kunming, Fuming County, China, in September 2013 and identified by Prof. J.-H. Kim, Gachon University, Republic of Korea. A voucher specimen (No. DiAB-2006-141) was deposited in the Herbarium of Korea Institute of Oriental Medicine, Republic of Korea. Dried leaves and twigs of *O. schwerinae* (1.0 kg) extract and its major compounds [2′-*O*-acetylvitexin (4.2 mg, *t*_R_ 18.3 min), quercitrin (12 mg, *t*_R_ 22.8 min) and hyperoside (7 mg, *t*_R_ 28.5 min)] were further purified, as previously described [15]. To check the its major compounds, high-performance liquid chromatography (HPLC) analysis was performed using an Agilent 1200 HPLC instrument chromatogram of OSSC. The column temperature was maintained at 30 °C. The analysis was performed at a flow rate of 1.0 mL/min and monitored at UV254 nm [15].

### 4.2. Cell Culture

The ARPE-19 cells were purchased from the American Type Culture Collection (ATCC CRL-2302; Manassas, VA, USA) and maintained at 37 °C in a humidified 5% CO_2_ incubator, as previously described [25].

### 4.3. Determination of Apoptosis Using Flow Cytometry

The rate of apoptosis was examined by flow cytometry using an annexin V-fluorescein isothiocyanate (FITC) and propidium iodide (PI) apoptosis detection kit (BD Bioscience, San Jose, CA, USA) following the manufacturer’s protocol. ARPE-19 cells were placed into 6-well plates at the density of 2.5 × 10^5^ cells/well were treated with 1 mM MG in the presence or absence of OSSC extract and its major compound for 24 h. Following digestion with trypsin and washing twice with PBS, the cells were labeled with 5 μL Annexin V-FITC and 5 μL PI for 5 min each at room temperature for 15 min in the dark. After incubation, 400 mL binding buffer was added and the percentage of apoptotic cells was analyzed using a FACSCalibur flow cytometer (BD Biosciences, Franklin Lakes, NJ, USA).

### 4.4. Western Blot Analysis

Polyacrylamide gel electrophoresis was performed, as described previously [25]. Membranes were probed with polyclonal antibodies against phosphorylated p53 (p-p53; Santa Cruz, CA, USA), Bax and Bcl-2 (Cell Signaling, Beverly, MA, USA), each at a 1:1000 dilution. The signals were detected using a WEST-one ECL solution (Intron, Korea) and captured on Fuji film LAS-3000 (Tokyo, Japan).

### 4.5. Statistical Analysis

Data are expressed as the mean ± S.E.M. ANOVA with Tukey’s test was used for multiple comparisons using Prism 5.04 software (GraphPad, San Diego, CA, USA).

## 5. Conclusions

In conclusion, we measured the antiapoptotic activities of the OSSC extract (0.1 μg/mL) and its three major compounds (0.01 μM) on MG-treated ARPE-19 cells, finding that the expression of the apoptotic factors p-p53, Bax and Bcl-2 was attenuated. Taken together, these results suggest that the use of the OSSC extract and its major compounds could prevent the development of DR.

## 6. Patents

The patents related to this study were registered in Kore (No. 10-097394), China (No. XL200980160639.3), Hong Kong (No. HK1170958), England, France, Swiss, Germany (No. 247483), the United Arab Emirates (No. 1028) and the USA (No. 8784,911).

## Figures and Tables

**Figure 1 molecules-25-02605-f001:**
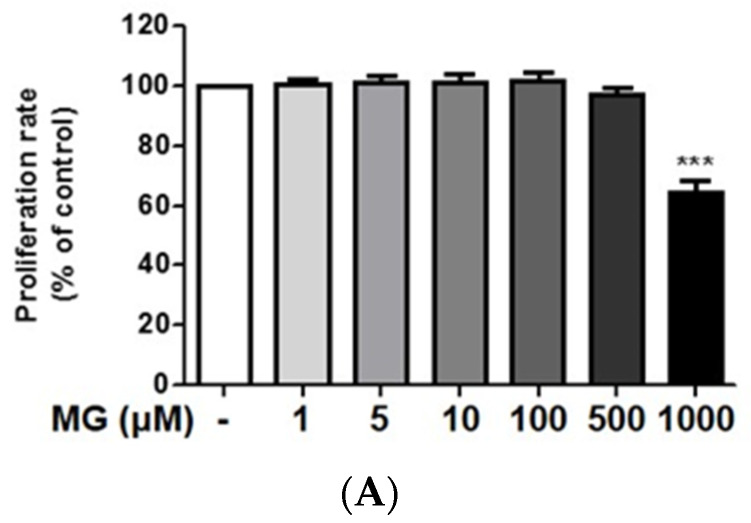
Effects of the *Osteomeles schwerinae* C. K. Schneid. (OSSC) extract and its major compounds on the viability of methylglyoxal (MG)-treated cells; (**A**) proliferation rates. Data are representative of three independent experiments and are expressed as the mean ± S.E.M. (*n* = 4). *** *P* < 0.01 vs. control; (**B**) FACS analysis. *** *P* < 0.01, ** *P* < 0.05 vs. control, respectively; (**C**) proliferation rates after treatment with the OSSC extract (0.1−20 μg/mL) and its major compounds (0.01–1 μM). *** *P* < 0.001 vs. control; ^###^
*P* < 0.001, ^##^
*P* < 0.01, ^#^
*P* < 0.05 vs. MG, respectively.

**Figure 2 molecules-25-02605-f002:**
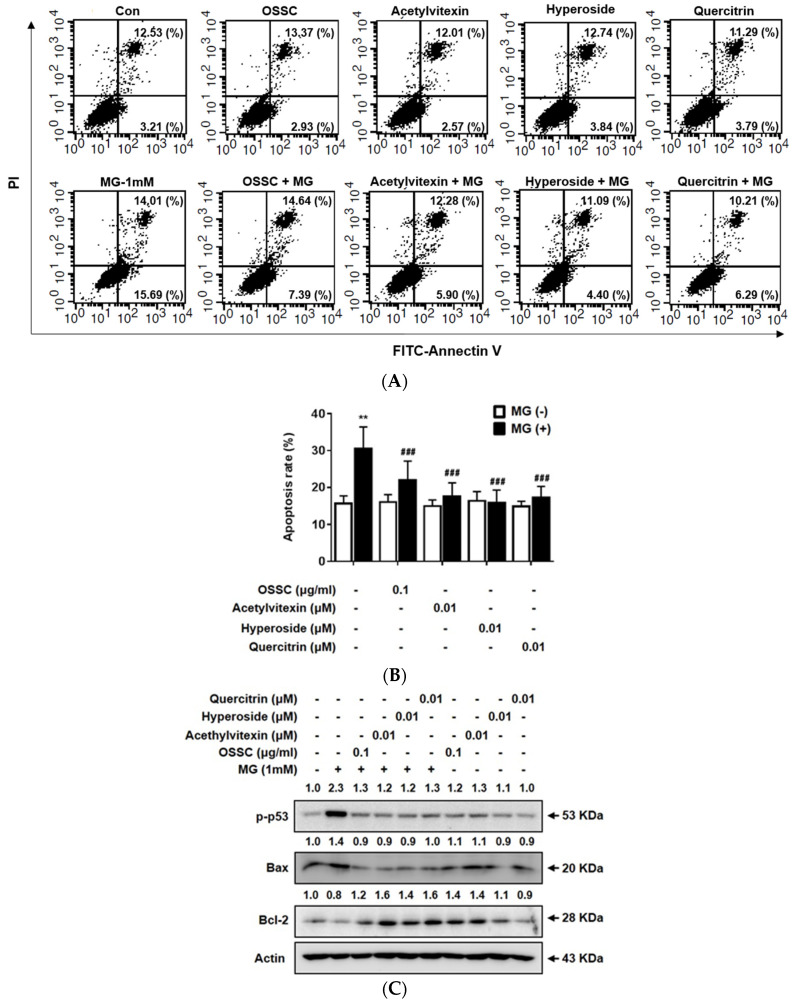
Effects of the OSSC extract and its maker compounds on MG-induced apoptosis. (**A**) Annexin V and PI staining; (**B**) percent rate of apoptosis. Data are expressed as the mean ± S.E.M. (*n* = 3). ** *P* < 0.01 vs. control; ^###^
*P* < 0.001 vs. MG; (**C**) expression of apoptosis-related proteins (p-p53, Bax and Bcl-2). Actin as the loading control.

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
