# Peer review of "Osteomeles schwerinae* Extract and Its Major Compounds Inhibit Methylglyoxal-Induced Apoptosis in Human Retinal Pigment Epithelial Cells"

_molecules, 2020, doi:10.3390/molecules25112605_

Round 1

Reviewer 1 Report

Osteomeles schwerinae C. K. Schneid is a wild species not well know so could be of interest to describe better the plant and to provide additional information about the origin of the vegetal material utilized and if the seed of this species has been provided by a genebank (indicate the accession code) or if the material samples were collected in the wildside and in this case is requested toindicate the geographical coordinates. 

Reviewer 2 Report

Dr. Pyun and colleagues, investigated the role of OSSC extracts in mitigating MG-related apoptosis in ARPE19 cells under toxic conditions. Overall, the manuscript is well written, conclusions are supported by results shown, however I have the following comments.

1] methods should include HPLC analysis.

2] What were the MG and OSSC extracts diluted/ dissolved in? indicate the reason for using this dilutant and concentration used.

3] Fig2A is "cropped" on the right, please replace.

4] Indicate full name for ROS and MG, before they are introduced in the results sections.

5] FACS plots are missing "X and Y axis" legends.

6] What is "control" in FACS figures 1A and 2A. Vehicle control maybe?

Reviewer 3 Report

The reviewed article is well prepared in terms of editorial and editing (although the quality of Figures is not at the highest level).

However, I have a few serious doubts as to the good reason of its publication in the Molecules Journal. The causes for this are as follows:

  1. As the Authors themselves indicate (or maybe the Editorial Team does), this article has the status of a Communication. I think that with its substantive and experimental scope, it is still exaggerated positioning.
  2. The only novelty element of the article is the use of the new system of plant extract with respective enzyme for the determination of apoptosis. All the rest are well-known raw materials and methods previously described by the same or similar team of Authors.
  3. The Authors quoted in the reference list some of their published papers very closely related and similar to the one currently sent to the Molecules Editorial Office, but not all substantially related to the subject (e.g. Evidence Based Compl. Altern. Med. 2018 ID6824215).
  4. The thematic similarity of this work and previous ones from a similar scope is too great for the described research to be treated as a separate article.
  5. The scientific research scope of this work is too small even for a communication.
  6. Use methodology of only one biochemical system is described in the paper.

And now, substantive remarks:

  1. It is generally accepted that when an extract is obtaining from a plant, its anatomical part used to produce this extract is given as a part of extract name. In this work, the phrase "Osteomeles schwerinae extract" is repeatedly used without the exact element of the plant being specified. And only in one, not significant place, reader can read that this extract is from leaves and twigs (line 59).
  2. I am not convinced that the three compounds (acetylvitexin, hyperoside and quercitrin) can be called "major compounds" in this plant. This is indicated by their quoted quantity in lines 60-61.
  3. The term "maker compounds" is sometimes used in place of "major compounds". Such a change is possible for term acetylvitexin, but hyperoside and quercitrin are definitely not "maker compounds" because they are common throughout the plant world.
  4. In the list of abbreviations it is required to respective organize of the recording system.

In summary, after minor corrections, this article is suitable for printing, but in other journal, more profiled for the study of activity of plants ingredients and preferably with the word "communications" in the title. In the extreme case, after correction, I suggest re-sending the work to the magazine "Plants" also edited by the MDPI, but with the need for its re-evaluation by reviewers.

Reviewer 4 Report

Pyun et al. prepared a manuscript on “Osteomeles schwerinae extract and its major compounds inhibit methylglyoxal-induced apoptosis in human retinal pigment epithelial cells”. Methodology used requires significant effort to obtain results. However, overall results ar not overwhelming.

Remarks:

  1. Authors should extend description of the plant, as it is encountered in Asia – many of readers are not familiar with it.
  2. Article is not organized per journal guidelines.
  3. Flow cytometry is should be described in detail as no reference is provided for the methodology.
  4. Figure 1A. Suppression of proliferation rate at 1 mM rate is not worth exploring further. Moreover, individual compounds are also used in mM rates. Thus, use of this plant extract does not seem viable option.
  5. Figure 2A. is incomplete.

Round 2

Reviewer 3 Report

I have read the authors' reply and the corrections in the text. I am inclined to agree with the presented argumentation, especially considering the opinions of other reviewers.

General remarks were also answered in a general way. However, I can find this convincing enough. My detailed suggestions for proposed changes in the text have been taken into account.

Now, after making the changes, I accept the publication of the text in its current form.

Reviewer 4 Report

Authors addressed all of the concerns raised.